# White Light-Photolysis for the Removal of Polycyclic Aromatic Hydrocarbons from Proximity Firefighting Protective Clothing

**DOI:** 10.3390/ijerph191610054

**Published:** 2022-08-15

**Authors:** Aline Marcelino Arouca, Victor Emmanuel Delfino Aleixo, Maurício Leite Vieira, Márcio Talhavini, Ingrid Távora Weber

**Affiliations:** 1Federal Institute of Education Science and Technology of Brasilia—IFB, Subcentro Leste—Complexo Boca da Mata, 02, Samambaia Sul, Brasilia 72302-300, Brazil; 2Laboratory of Inorganic and Materials (LIMA), Chemistry Institute, University of Brasilia—UNB, Brasilia 70904-970, Brazil; 3National Institute of Criminalistics, Brazilian Federal Police, SAIS Quadra 07 Lote 23, Brasilia 70610-200, Brazil

**Keywords:** photolysis, polycyclic aromatic hydrocarbons (PAHs), firefighters, personal protective equipment (PPE), proximity firefighting protective clothing (PFPC), advanced oxidation process (AOP)

## Abstract

The presence of polycyclic aromatic hydrocarbons (PAHs) on firefighters’ personal protective equipment is a concern. One form of preventing from these compounds is to decontaminate proximity firefighting protective clothing (PFPC). Traditional decontamination methods do not promote total removal of pollutants and alter the properties of PFPC. The objective of this work was to evaluate the effectiveness of white light-photolysis (WLP), an advanced oxidation process (AOP), for removing PAHs from PFPC, while maintaining the integrity of the fabric fibers. Experiments were carried out, varying reaction time and concentration of H_2_O_2_. With WLP (without H_2_O_2_), it was possible to remove more than 73% of the PAHs tested from the outer layer of PFPC in 3 days. The WLP provided the greatest removal of PAHs, compared with the most common mechanical decontamination techniques (laundering and wet-soap brushing). The fibers’ integrity after exposure to the white light was evaluated with infrared spectroscopy and scanning electron microscopy/energy dispersive X-ray spectrometry. In addition, a tearing strength test was performed. No remarkable fabric degradation was observed, indicating a possible, routine-compatible, simple, and inexpensive method of decontamination of PFPC, based on photolysis, which is effective in the degradation of PAHs and maintains the integrity of fabric fibers.

## 1. Introduction

Firefighting is a high-risk activity, affecting the physical and mental health of these professionals. The International Agency for Research on Cancer (IARC) has classified firefighters’ exposure to toxic materials as Group 2B, that is, possibly carcinogenic to humans [1]. As well as the obvious effects of combustion of hazardous materials, the contamination of firefighters’ proximity protective clothing may promote danger to their health as well.

One of the main compounds that is formed during the combustion and pyrolysis process of a material are the polycyclic aromatic hydrocarbons (PAHs) [2,3]. PAHs represent a class of complex organic chemicals with more than 100 compounds containing two or more aromatic rings condensed in different ways in their structures [4]. PAHs can be formed by different pathways, as in an oil spill or oil seepage. The incomplete combustion of organic matter and fossil fuels is the most prominent source of PAHs in the environment [5]. The concern about the presence of these compounds in fire residues is due to the fact that many PAHs are considered carcinogenic, mutagenic, and teratogenic [6,7].

In the case of firefighters, the main routes of exposure are by inhalation and dermal absorption. One way to reduce these contaminations is through the use of a self-contained breathing apparatus (SCBA) and the use of proximity firefighting protective clothing (PFPC) [8]. For dermal contamination, exposure can occur through the deposit of the contaminants directly on the skin or through cross-contamination. This can be evidenced by the fact that even when full personal protective equipment (PPE) is being used, including SCBA, the PAHs level in firefighters is higher in post-combat situations than pre-combat situations [8,9,10]. This is due to the cross-contamination that happens after the firefighters’ contact with a contaminated suit or equipment [11].

In order to reduce cross-contamination, an effective decontamination of all personal protective equipment must be carried out. A number of studies have been carried out to develop a safe and effective method of mechanical decontamination of proximity firefighting protective clothing. Most decontamination (decon) methods involve laundering, brushing, and using compressed air. Some of these mechanical decon techniques are not sufficient to remove PAHs completely [12,13,14,15]. Another important fact is that the PFPC is an expensive piece of equipment. In a 2017 bid made by the Military Firefighting Corps (Corpo de Bombeiros Militar) of the Federal district of Brasilia, the purchase price of a unit cost €1419.58 [16]. Considering this high purchase price per unit, suit conservation is extremely important, so any decon method must not damage it. Laundering, however, may alter the properties of fabric of the firefighting turnout gear [17,18,19].

An alternative to mechanical decon is chemical decontamination, such as advanced oxidation processes (AOPs). Lucena et al. [20] evaluated the ozonolysis decon of model PAHs (pyrene and 9-methylanthracene) in pieces of impregnated turnout firefighters’ fabrics. Despite being able to partially decontaminate the sample, the result was unsatisfactory (due to the low degradation rate obtained and to the formation of compounds with similar or greater toxicity than parent compounds, such as phenanthrene-derived compounds).

Considering positive results of using photolysis, an effective and less expensive AOP used in the degradation of PAHs in liquid [21,22,23,24] and/or solid [25,26,27] samples, this represents a possible way to decontaminate firefighters’ suits. This approach is favored because most PAHs absorb in the range of 210–386 nm [28,29]. In addition, photolysis is a simple, easy-to-perform, and low-cost method which a priori tends to not damage fabrics.

The goal of this paper is to evaluate the AOP chemical decon of firefighting protective clothing impregnated with PAHs using white light photolysis as an alternative to the mechanical decon methods already used. Besides studying the effect of reaction time on PAHs removal, we also evaluated the addition of hydrogen peroxide, an oxidizing agent that favors radical oxidation reactions. The aim of the study was to maximize the removal of polycyclic aromatic hydrocarbon pollutants while minimizing damage to the fabric fibers, and to maintain integrity of the fibers as well as the physical properties of the turnout gear.

## 2. Materials and Methods

### 2.1. Chemicals and Reagents

The analytes 9-methylanthracene (9MA, 98%) and pyrene (PYR, 98%) were purchased from Sigma-Aldrich (São Paulo, Brazil) and used without additional purification. Acetonitrile, suitable for HPLC (>99.9%, Exodo and JT Baker, São Paulo, Brazil), hydrogen peroxide (H_2_O_2_ 35% *w*/*v*, Synth, São Paulo, Brazil), and deionized water (≥10 MΩ cm^−1^), produced in a Milli-Q purification system, were used in all experiments.

All glassware was washed with a neutral detergent, and rinsed with deionized water, then re-rinsed with ethanol (Synth), ethyl acetate (Synth), and dichloromethane (Merk, São Paulo, Brazil), all HPLC grade.

### 2.2. White Light-Photolysis Experiments

All experiments were performed in a photoreactor equipped with a 250 W mercury vapor and tungsten filament white light (emission spectrum in Appendix A), and two coolers to promote cooling and airflow, as shown in Appendix A.

PAHs (9-methylanthracene and pyrene) were deposited onto a piece of fabric from the outer shell of the firefighting suits. The fabric had a nominal composition of 58% para-aramid (Kevlar^®^), 40% meta-aramid (Nomex^®^), and 2% carbon from Unishell^®^ with a water repellant coating of fluorocarbon (Teflon^®^). The samples of fabric were provided by Santanense Workwear and were used in the manufacturing of Brazilian firefighters’ protective clothing. Pieces of fabric were cut into 3 × 3 cm^2^ squares (Appendix A).

A 0.005 M pyrene and 9-methylanthracene stock solution was prepared in acetonitrile. Then, 200 μL was dripped onto the fabric samples. The solvent solution was air-dried for 10 min at room temperature. After 10 min, 200 μL of H_2_O_2_ was also deposited onto the fabric surface containing PAHs. H_2_O_2_ concentrations of 0%, 0.35%, and 3.5% of H_2_O_2_ in Milli-Q water were evaluated. Photolysis (WLP) was carried out inside the photoreactor for periods of 0, 1, and 3 days. A similar experiment was conducted without light irradiation (BLK), as reference.

After the photolysis reaction took place, the fabric samples were extracted with 10 mL of acetonitrile in a test tube. Each tube was placed into an ultrasonic bath for 20 min at room temperature. After this procedure, the extracts were stored in amber flasks and stored in a freezer. The extracts were analyzed by UV/VIS spectroscopy.

The kinetics were evaluated. Tests were conducted on the best condition of white light-photolysis reaction (0% of H_2_O_2_) and in different reaction times (0, 1, 3, 6, 9, 12, and 15 days). The extracts were obtained as previously described and stored in a freezer for further analysis. These samples were analyzed with a gas chromatogram coupled with mass spectrometer (GC/MS) in order to determine the PAHs concentration and to detect possible photolysis by-products.

### 2.3. Three-Layer Fabric White Light-Photolysis Decontamination

In order to analyze the effect of white light-photolysis decon within a more realistic scenario, the decontamination procedure was carried out with real samples of the three-layer fabric from used Brazilian firefighters’ protective clothing.

The fabrics from the turnout gear were cut into 10 cm diameter disks and sewn together in the same order that is found on the Brazilian firefighters’ gear (Appendix A). Each disk was contaminated with 1.8 mL of the PAHs stock solution, then was air-dried for 10 min. The best reaction conditions found in the previous tests (reaction time and quantity of H_2_O_2_) were reproduced. For the PAHs extraction, 80 mL of acetonitrile was used with the fabric disks in an ultrasonic bath for 20 min at room temperature.

### 2.4. Three-Layer Fabric Mechanical Decontamination

To compare the white light-photolysis decon with mechanical decontamination techniques, a series of experiments were carried out on the three-layer fabric samples.

For the wet-soap brushing decon, the procedure described by Fent, K.W., et al. [13] was adapted. A neutral soap (0.5 mL of soap in 380 mL of water) solution was sprayed onto a pre-soaked three-layer sample using a spray bottle. Then, the disk was scrubbed 10 times with a plastic bristle brush and rinsed quickly, preventing water from penetrating the bottom layer of the sample.

Another common decon procedure which was evaluated was laundering. To evaluate the efficiency of this technique, an experiment using a bucket and a mechanical stirrer was carried out, simulating a washing machine. This adaptation was necessary given the small size (10 cm) of the disks (instead the whole suit).

The simulated laundering decon was adapted following the methodology described by NFPA 1851 [30], described in the Appendix A. Two washing cycles (20 min then 10 min) were completed, using 1 mL of a commercial liquid laundry detergent. Then, three rinse cycles were completed: one for 10 min and two more for 5 min each. In all cycles, 4 L of water was used, and after each cycle time the water was completely drained.

### 2.5. Extract Analysis

#### 2.5.1. UV-VIS Spectroscopy

After the decontamination experiments, absorption spectra of extracts (Varian ultraviolet/visible spectrophotometer, model Cary 5000) were obtained to determine the concentration of each target compound. For this, the extract was diluted in a volume of 1:1 (outer shell fabric experiments) and 1:2 (three-layer fabric experiments) for better analysis. The spectra were obtained in the 220–400 nm range.

Quantification of the analytes was conducted with an analytical curve (9-methylanthracene and pyrene). The curve and the ANOVA table are shown in Appendix A. The absorbance of samples was subtracted from blank samples (without the PAHs) for the concentration determination.

#### 2.5.2. Gas Chromatography-Mass Spectrometry

The concentration of pyrene and 9-methylanthracene was determined by gas chromatograph (Agilent model 6890N) coupled with mass spectrometer (Agilent model 5973 inert), in samples from the kinetics evaluation. The ASTM method 8270E [31] was used with a Rxi^®^-1 ms stationary phase capillary column, with 100% methylpolysiloxane, dimensions 25 m × 0.20 mm × 0.33 µm (RESTEK).

The chromatographic operating conditions were the following. The injector temperature was maintained at 280 °C, in Splitless mode with 1.3 μL injection. The column was maintained with a constant flow of helium at 0.6 mL.min^−1^. The chromatographic oven programming had an initial temperature of 40 °C, held for 4 min, heating at a rate of 10 °C.min^−1^ to 320 °C, and remaining at this temperature for 2 min. The total time of analysis was 34 min. Solvent delay was used for 4.00 min and a gain factor of 20.

The GC/MS interface was maintained at 280 °C, and the mass spectrometer was operated in scan mode in the scan range from 35 to 500 *m*/*z*, with HiSense.u. The mass spectra obtained were analyzed using the Chemstation Data Analysis program and the NIST Search program (version 2.3). Quantification was conducted with external analytical curves of 9-methylanthracene and pyrene. All the data is available in the Appendix A.

### 2.6. Fabric Evaluation

To assess whether the white light-photolysis decon would deteriorate the outer shell fabric used in firefighting protective clothing, Unishell^®^ samples were analyzed. Samples of fabric without PAHs previous contamination were exposed for 3 days to white light (WLP) and without it (BLK), using 0%, 0.35%, and 3.5% H_2_O_2_. Additionally, a long exposure (30 days) test was performed with the best white light-photolysis condition.

#### 2.6.1. Attenuated Total Reflectance Fourier Transform Infrared (ATR-FTIR) Spectroscopy

Infrared analysis was carried out using a Thermo Scientific Nicolet iS10 FTIR Spectrometer with The Thermo Scientific Smart iTX ATR sampling accessory. Spectra were collected in the range of 460–4000 cm^−1^ with a DTGS detector, KBr beam splitter, and a HeNe laser.

For each fabric sample, 10 FTIR spectra were collected at different locations on the sample and the average spectrum was obtained. Spectral baseline correction and normalization was carried out using Origin 2021 program.

#### 2.6.2. Tearing Strength

The sample submitted to photolysis best condition for 30 days (long exposure) was chosen to be submitted to a tear resistance test. This was performed in accordance with ASTM D2261:2013 (2017)e1 [32], Standard Test Method for Tearing Strength of Fabrics by the Tongue (Single Rip) Procedure (Constant-Rate-of-Extension Tensile Testing Machine). The experimental conditions are described in the Table 1 below.

#### 2.6.3. Scanning Electron Microscopy with Energy Dispersive X-ray Spectrometry SEM/EDS

SEM/EDS images were obtained from fabric fibers after the treatment using a Zeiss, model EVO 15, Scanning Electron Microscope with EDS Oxford UltimMax 40. Fiber diameters were obtained using the Gwyddion^®^ 2.60 program: 10 points were measured in 5 images for each sample (total of 50 measurements/sample). The images were obtained using a backscattered electron detector and low vacuum, without conducting coating. For the compositional maps, a 40 mm square silicon detector was used and AzTech software.

### 2.7. Statistical Evaluation

All experiments were conducted in triplicate (*n* = 3). All data obtained were evaluated using ANOVA test and paired-sample *t* test to assess the statistically significant differences among mean values.

## 3. Results and Discussion

### 3.1. White Light-Photolysis Experiments

The spectrum of the white lamp has several bands, both in lower wavelengths (283 nm, 322 nm, 353 nm) and in the longer wavelength region (465 nm, 496 nm, 498 nm, 545 nm, and 577 nm). Comparing the absorption spectra of the model PAHs with emission spectra, we can see a superposition of spectra, indicating that this lamp was suitable for the PAHs photolysis. All spectra are available in Appendix A.

Before analysis of the effect of H_2_O_2_ concentration and reaction time, an experiment was conducted in the dark (BLK) without using any energy source. After 3 days, the longest reaction time tested, approximately 20 ± 1% of PAHs removal had been obtained (Figure 1a). This removal is probably related to the carryover of the analytes due to the airflow maintained inside the photoreactor and due to the compound volatilization. The addition of an oxidizing agent (H_2_O_2_,) did not affect PAHs concentration (Figure 1b,c). About 23 ± 6% and 26 ± 8%of PAHs removal was obtained with 0.35% and 3.5% of H_2_O_2_, respectively. This may be related to the fact that hydroxyl radical formation is favored in an aqueous medium and light is necessary to break the O-O bond in H_2_O_2_ [24].

Determining the basal level of PAHs removal, the same experiment was conducted using the white light as an energy source. Again, the H_2_O_2_ content varied from 0 to 3.5% *w*/*v*. The best result was obtained without H_2_O_2_ (0% H_2_O_2_), as shown in Figure 1d. As observed in the BLK experiments, there was no change by adding an oxidizing agent. About 65 ± 9% and 83 ± 12% of removal was obtained with 1 and 3 days of light exposition. With the addition of 0.35 or 3.5% of H_2_O_2_, very similar results were achieved (81 ± 8% and 76 ±5%, respectively, removal after 3 days). This result strongly suggests that PAHs tested are more susceptible to photoinduced degradation than to hydroxyl radical attacks. This is not surprising, as the reaction was not carried out in an aqueous media (in which analytes are insoluble). Moreover, considering potential applications, this can be seen as quite an encouraging result, since very mild conditions were used. Comparing the PAHs removal (Figure 2), the best condition was determined as 3 days of irradiation of with light without H_2_O_2_ (about 83% removal of PAHs).

### 3.2. White Light-Photolysis Kinetics

For determination of the degradation kinetics of each PAHs, the samples were submitted to longer exposure and the extracts obtained after the WLP (without H_2_O_2_) were analyzed by GC/MS. Both 9-methylanthracene and pyrene follow a pseudo-second-order kinetics. The plot of 1/[PAH] versus reaction time was used to determine the rate constant K (Figure 3).

The 9-methylanthracene curve draws attention for presenting two regions: initially a faster reaction, with a rate constant of 53.8, and then a slower kinetic, with K = 3.8. Pyrene, on the other hand, presents a well-behaved curve with a unique constant rate K = 11.67. The efficiencies obtained throughout the test are described in Figure 4.

For 9MA, given that the first step reaction was relatively fast, 70% of the compound was removed after only one day. Considering the cost–benefit, this implies that the photolysis decon method could be performed in one day. With the slower kinetics of PRY, on the other hand, relevant removal is observed around the 6th day of exposure. The optimal results, considering both PAHs, were obtained between 6th and 9th days. In real situations, however, this exposure time is very long and works against optimization of the cost–benefit ratio.

Additionally, the mass spectra obtained in this experiment were evaluated and no PAHs derivatives more harmful than the original PAHs were formed.

### 3.3. Evaluation of Fiber Integrity

In order to assess fiber integrity after photolysis, Fourier Transform Infrared/Attenuated Total Reflection (FTIR/ATR) spectra and SEM/EDS images were acquired, and a tearing strength test was carried out.

#### 3.3.1. FTIR/ATR

The fabric contained a mixture of Nomex^®^ (meta-aramid) and Kevlar^®^ (para-aramid) fibers, which can undergo oxidation forming acids, alcohols, and/or amides [33,34,35]. In the case of fiber degradation, it was possible to observe a reduction in the intensity of the amide bands and an increase in the intensity of the bands in the O-H and N-H regions. FTIR spectrum of untreated fabric was compared to treated samples. The spectra are shown in Figure 5.

All the spectra are similar. No new band was observed nor even the widening or shifting of existing bands, suggesting that the fabric did not undergo chemical changes through treatment. The band at 3307 cm^−1^ refers to the N-H stretch of the amide of the para-aramid and meta-aramid compounds, whereas the band at 1724 cm^−1^ refers to the C=O stretch. The fibers have aromatic rings; it is possible to identify the C=C aromatic stretch at 1640 cm^−1^. The region at 1017–820 cm^−1^ corresponds to the out-of-plane vibration of the C-H, indicating a meta and para substitution. Bands in the 2918–2850 cm^−1^ range are characteristic of the C-H sp^3^ stretch. However, since fibers are not expected to have saturated carbons, the bands must have come from water repellant carbon-based coating, which seems to have been preserved during treatment. For the Unishell^®^ clothing, the outer shell fabric has a fluorocarbon (Teflon^®^) coating, which is consistent with the C-H sp^3^ stretch. The other attributions of the bands are described in Table 2 and are in accordance with what is described in the literature [33,35,36].

In addition to the spectra obtained after 3 days of decontamination, an extended experiment was carried out with WLP with 0% of H_2_O_2_ for 30 days (720 h). Even after the longer exposure (Figure 6), no changes in spectral profile were observed.

These results suggest the treatments based on photolysis degraded the deposited PAHs without altering the chemical structure of fibers. Even in prolonged experiments, no substantial differences were observed in the FTIR/ATR spectra.

Davis et al. [33] conducted a similar experiment, where outer shell fabrics used in firefighter jackets and pants were exposed to simulated ultraviolet sunlight at 50 °C and 50% relative humidity for different periods of time. The experiments were carried out in NIST SPHERE (Simulated Photodegradation via High Energy Radiant Exposure), a simulated photodegradation device. After 13 days exposure, the researchers observed a significant degradation of the water repellant coating. Additionally, changes in the fibers were identified by formation of compounds resulting from oxidation and break-up of the amides bonds. In our work, a lower energy irradiation source was used (white light), therefore there was less degradation of fabric than expected. In addition, the use of white light is cheaper and imposes less risk to the operator. The satisfactory result obtained using white light can be partially attributed to the match between the absorption of the PAHs and the emission of the used lamp.

#### 3.3.2. Tearing Strength

As described in the NIST report 1751 [34], a way to assess changes in fabric constitution and possible wear from the decontamination process is by performing a tearing strength test. The NFPA 1971 [37] standard, which determines the necessary conditions for firefighting protective clothing, indicates that the fabrics of the outer layer of combat suits must present tear resistance higher than 100 N.

For the tearing strength test, only the sample treated for 30 days with WLP and 0% H_2_O_2_ was compared to the untreated fabric. For the Unishell^®^ fabric, the tearing strengths were 260.97 ± 10.1 N and 218.39 ± 28.8 N for warp and weft, respectively. After the WLP decontamination process, tear strength values of 192.62 ± 31.8 N and 181.52 ± 22.9 N were obtained for warp and weft, respectively (Table 3).

There was a reduction of warp (26%) and weft (17%) values, which could be related to the breaking of the bonds in the fabric’s water repellant coating, and not necessarily due to fiber degradation, as described by Davis et al. [33]. This fiber treatment makes the fabric more resistant, increasing the force needed to tear it. By reducing the repellant coating there is a reduction in the tearing resistance. This fact indicates that there is a reduction in the fabric’s water repellant coating, not in the fabric’s fibers bonds, that is, no alterations were observed in the FTIR/ATR spectra that would indicate a break-up of the para-aramid and meta-aramid molecule.

#### 3.3.3. SEM/EDS

Micrographs of outer shell fabric were obtained to visualize the morphology of the fibers and investigate possible alterations. Images were obtained after decontamination for 3 and 30 days, as well as for the untreated sample, as shown in Figure 7.

No change in fabric morphology was perceived and all fibers showed similar diameters within the standard deviation (Table 4). The results suggest no damage on fabric surface and the fibers maintained their structure, even for those undergoing photolysis with white light for 30 days.

This result reinforces the conclusion that the reduction in tearing strength described in the previous item was associated with the Teflon^®^ water repellant coating. Teflon^®^ is a polytetrafluoroethylene (PTFE) polymer used as coating for various materials [38]. Due to the presence of a fluorine element, the presence of PTFE coating can be analyzed by scanning electron microscopy (SEM) with energy dispersive X-ray spectrometry (EDS). For this, elemental mapping of several points on untreated and treated fabric for 30 days with WLP 0% H_2_O_2_ by SEM/EDS were obtained. (Figure 8). The untreated sample showed a higher density of fluorine, indicating the presence of the protective PTFE coating layer. The 30 days treated sample showed a reduction in fluorine distribution, which indicates the partial removal of PTFE coating in the outer shell samples, corroborating the idea that polyamides remain intact after treatment and reduction in tearing strength can be attributed to PTFE partial removal. It is important to point out that PTFE coating removal is expected and occurs naturally with the use of the turnout gear and needs to be replaced after some period. All maps obtained are available in Appendix A.

### 3.4. Three-Layer Fabric Decontamination: Mechanical Decon vs. White Light-Photolysis Decon

To compare the WLP with traditional mechanical decon, pieces of three-layer fabric were contaminated with pyrene and 9-methylanthracene. The photolysis was carried out according to previous tests carried out for 3 days. Wet soap brushing and laundering was performed as described by Fent et al. [13].

After performing the tests, the samples were extracted with acetonitrile and the extracts analyzed by UV/VIS for PAHs quantification. Degradation rates and spectra related to each decontamination are depicted in Figure 9.

The simulated laundering reduced the PAHs contamination by 44 ± 12% of PAHs, while the wet-soap brushing reduced it by 32 ± 12%. Decontamination via photolysis gave the best result, 73 ± 7%. This result demonstrates that photolysis with white light is efficient in removing the PAHs deposited on firefighters’ protective clothing up to three layers of fabric.

One of the main complaints related to laundering and wet-soap brushing is the fact that the turnout gear remains damp after decontamination. This increases the risk of burns, in addition to the discomfort and extra weight [39] or the need to wait a drying interval before using the suit. The WLP, being a dry method, provides an advantage, in addition to being relatively fast and efficient.

Another advantage of WLP is that there is a lower risk of cross contamination. The firefighter ends up having less contact with the contaminated protective clothing when compared with mechanical decontamination methods. When laundering is done at home (the most common in Brazil), there is a greater risk of contaminating of other pieces of clothes and even other residents, due to the use of the washing machine.

Another important fact is the durability of the turnout gear. The washing process can promote further fiber degradation. Horn et al. [17] identified a reduction in tearing strength tests after laundering. The group evaluated the three layers of fabric and three methods of decontamination: laundering in a washing machine, wet-soap brushing, and dry brushing. Laundering had lower trap tear strength than the other treatments evaluated. After 10 washes, there was a reduction of 1.2–9.1%. With 20 washes the reduction was 9.3–19.6%, and with 40 washes it was 24.1–41.1%. Moreover, when fabric was submitted to 40 times in laundering machine, tear strength dropped below 100 N, the minimum requirement following NFPA 1971 [37].

The results discussed above could have been affected by experimental details and, therefore, present discrepancies between studies conducted by different groups. In this study, laundering was conducted on a small scale, using an adapted experimental set up to simulate a washing machine. The bucket adaptation does not promote the friction that facilitates washing, generated in the conventional washing machine. This may have disfavored PAHs removal and caused the lowest decontamination rate. Some studies have published more efficient results using mechanical decon methods in the removal of analytes than those described in this work.

Fent et al. [13] obtained wet-soap decon as the most effective technique in reducing PAHs contamination, with an 85% reduction. It is important to note that those authors did not use any chemical means for decontamination. In another study, performed by Banks, et al. [12], laundering decon method failed to promote the total removal of the PAHs and other compounds deposited on firefighting clothing after a fire event. In fact, in some cases, they found a higher concentration, indication a cross-contamination.

## 4. Conclusions

The goal of this research was to promote a chemical method for decontamination of polycyclic aromatic hydrocarbons deposited on firefighting turnout gear, using white light photolysis. The method provides an alternative to mechanical decon processes already used by firefighters.

The WLP decon promoted the removal of pyrene and 9-methylanthracene analytes deposited on fabrics of the outer layer of firefighter turnout gear without the need to add an oxidizing agent. After 3 days, the method promoted the removal of 81 ± 8% of both PAHs following a pseudo-second-order rate. The proposed decontamination route was compared with commonly applied techniques, laundering and wet-soap brushing, on three-layer fabric. Photolysis showed a removal of 73% of the deposited PAHs, while laundering removed 44% and wet-soap brushing 32%.

Regarding the fabric analysis, the protective clothing outer shell fabrics submitted to 3 and 30 days of photolysis were evaluated to assess whether the white light-photolysis decon promoted deterioration of the fibers. The chemical structure of the fibers was evaluated with FTIR/ATR; no changes were obtained in the spectra that could indicate the breakdown of fiber polymers. The tearing strength test presented a reduction in tearing strength by 17–26% after 30 days of light exposure. This reduction is not necessarily due to fiber degradation but related to the partial removal of fabric’s water repellant coating. The treated sample showed a reduction on fluorine density, indicating the suggesting that the Teflon was degraded, but still was present. No change was observed in the fibers’ morphology.

In view of these results, the 3 days WLP without H_2_O_2_ can be considered an efficient chemical decon method for the removal of PAHs deposited on proximity firefighting protective clothing, maintaining fabric’s integrity and properties. The WLP decon has several advantages in comparison to what is currently used. Besides promoting greater removal of PAHs than the mechanical decon techniques, it is not necessary with WLP to use water on the turnout gear, which can result in burns if the suit is used while still damp. In addition, the technique developed is simple, cheap, environmentally friendly, and safer for the firefighter, as it reduces contact with contaminants.

## Figures and Tables

**Figure 1 ijerph-19-10054-f001:**
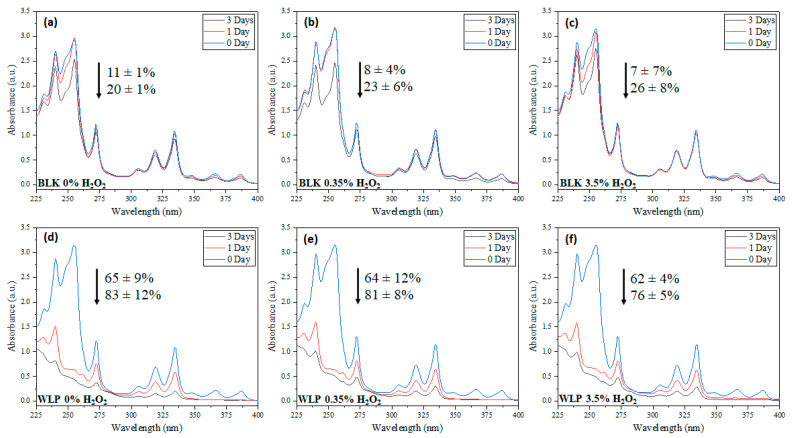
UV/VIS spectra of experiments performed in 1 and 3 days with (**a**) BLK 0% of H_2_O_2_, (**b**) BLK 0.35% of H_2_O_2_, (**c**) BLK 3.5% of H_2_O_2_, (**d**) WLP 0% of H_2_O_2_, (**e**) WLP 0.35% of H_2_O_2_, and (**f**) WLP 3.5% of H_2_O_2_.

**Figure 2 ijerph-19-10054-f002:**
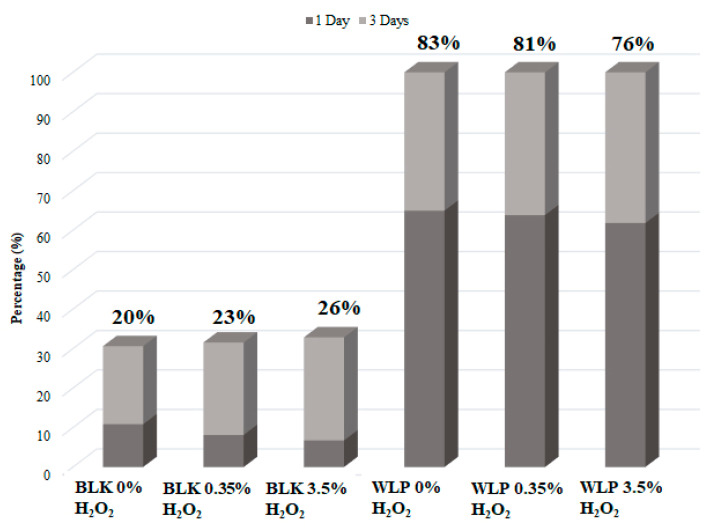
Comparative graph of PAHs removal in all tested conditions: BLK 0% of H_2_O_2_, BLK 0.35% of H_2_O_2_, BLK 3.5% of H_2_O_2_, WLP 0% of H_2_O_2_, WLP 0.35% of H_2_O_2_, and WLP 3.5% of H_2_O_2_.

**Figure 3 ijerph-19-10054-f003:**
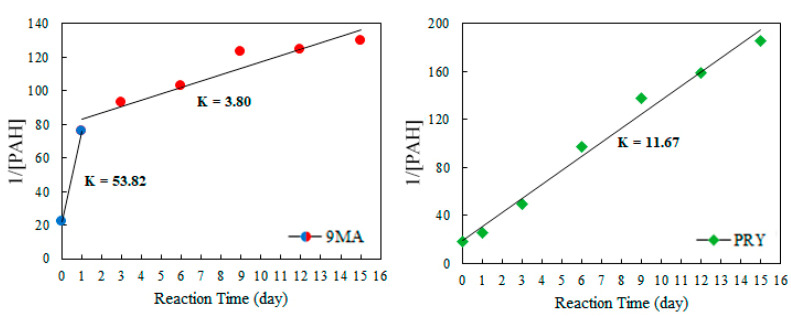
Pseudo second order rate constants for WLP (with 0% of H_2_O_2_) degradation reaction of 9-methylanthracene and pyrene.

**Figure 4 ijerph-19-10054-f004:**
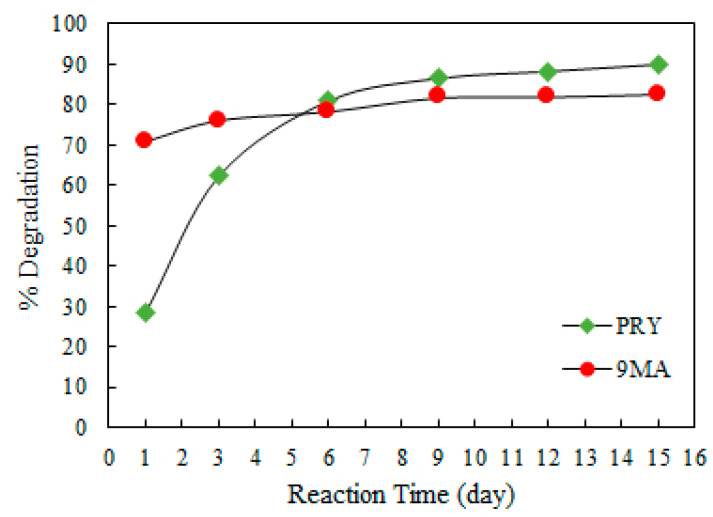
WLP (0% of H_2_O_2_) efficiency in degradation of PAHs obtained in 0, 1, 3, 6, 9, 12, and 15 days.

**Figure 5 ijerph-19-10054-f005:**
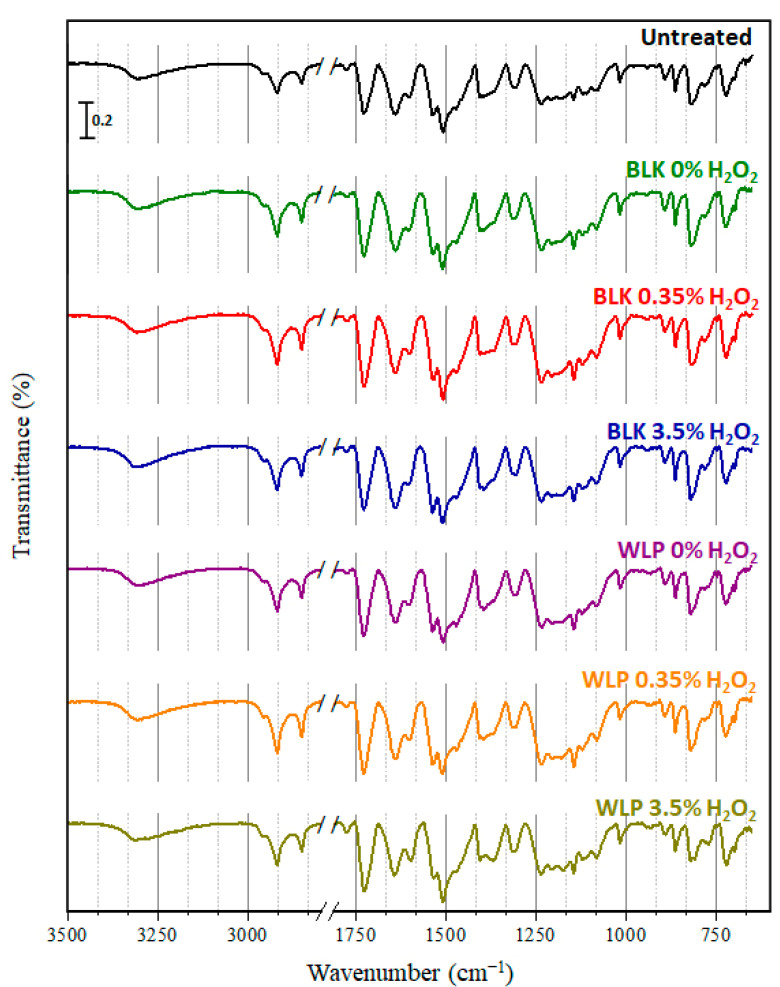
FTIR/RTA spectra of untreated sample and treated samples after exposure for 3 days: BLK 0% of H_2_O_2_, BLK 0.35% of H_2_O_2_, BLK 3.5% of H_2_O_2_, WLP 0% of H_2_O_2_, WLP 0.35% of H_2_O_2_, and WLP 3.5% of H_2_O_2_.

**Figure 6 ijerph-19-10054-f006:**
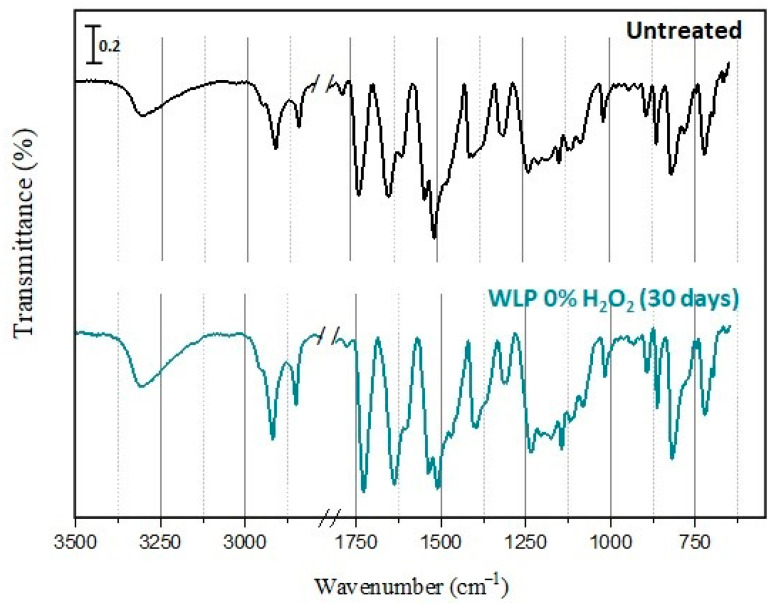
FTIR/ATR spectra of untreated sample and WLP 0% of H_2_O_2_ exposed for 30 days.

**Figure 7 ijerph-19-10054-f007:**
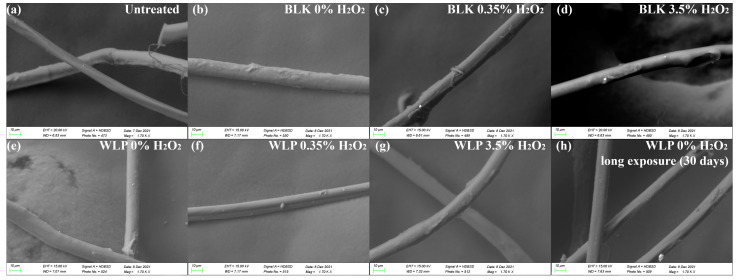
Micrographs of fibers before and after treatment with (**a**) untreated sample, (**b**) BLK 0% of H_2_O_2_, (**c**) BLK 0.35% of H_2_O_2_, (**d**) BLK 3.5% of H_2_O_2_, (**e**) WLP 0% of H_2_O_2_, (**f**) WLP 0.35% of H_2_O_2_, (**g**) WLP 3.5% of H_2_O_2_, and (**h**) WLP 0% of H_2_O_2_ for 30 days.

**Figure 8 ijerph-19-10054-f008:**
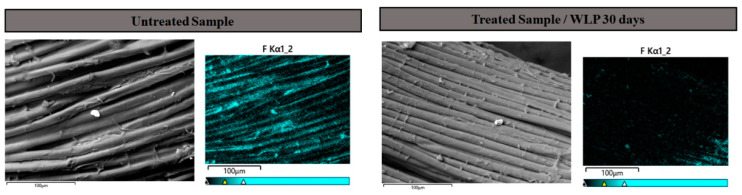
Compositional maps for fluorine for the untreated sample and for the treated sample with WLP and 0% H_2_O_2_ for 30 days.

**Figure 9 ijerph-19-10054-f009:**
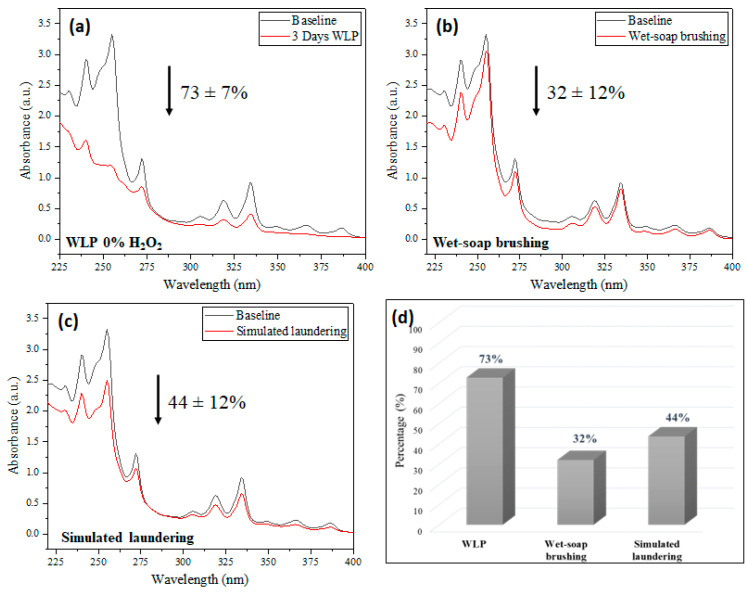
UV/VIS spectra of decontamination procedure performed by (**a**) WLP for 3 days without H_2_O_2_; (**b**) simulated laundering; and (**c**) wet-soap brushing. (**d**) Summary of percentage of PAHs removal.

**Table 1 ijerph-19-10054-t001:** Tearing strength experiment conditions.

Distance between the Jaws	75 mm
Speed	50 mm/min
Dynamometer Type	CRT
Calculation Methodology	Average of 5 peaks
Software Used	Bluehill 3
Dimensions of each jaw	Front: 2.5 mm × 7.5 mm Back: 2.5 mm × 7.5 mm
Tear Direction	Parallel to the warp and to the weft

**Table 2 ijerph-19-10054-t002:** Assignment of FTIR/ATR bands.

Band (cm^−1^)	Assignment
3307	N-H stretch
2918–2850	C-H sp^3^ stretch
1724	C=O amide stretch
1640	C=C aromatic stretch
1537–1472	N-H deformation in plane and C-N stretch
1411 and 1304	C-N aromatic stretch
1017–820	C-H vibration out-of-plane

**Table 3 ijerph-19-10054-t003:** Tearing strength (N) values with expanded uncertainty with 95% confidence, performed in accordance with ASTM D2261 [32].

	Warp	Weft
Untreated sample of fabric	260.97 ± 10.1	218.39 ± 28.8
After 30 days WLP decon	192.62 ± 31.8	181.52 ± 22.9
ΔN (absolute)	68.35	36.87
ΔN (%)	26%	17%

**Table 4 ijerph-19-10054-t004:** Fiber diameter.

Sample	Mean (μm)	Standard Deviation (μm)
Untreated	14.15	2.71
BLK 0%	13.90	1.50
BLK 0.35%	13.68	1.19
BLK 3.5%	14.35	1.62
WLP 0%	14.00	2.57
WLP 0.35%	14.36	1.26
WLP 3.5%	13.85	1.95
WLP 0% for 30 days	14.53	1.80

## Data Availability

Please, find the Appendix A. For the moment, no additional datasets are publicly available online.

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
