# Peer review of "White Light-Photolysis for the Removal of Polycyclic Aromatic Hydrocarbons from Proximity Firefighting Protective Clothing"

_ijerph, 2022, doi:10.3390/ijerph191610054_

Round 1

Reviewer 1 Report

Manuscript written by Aline M. Arouca et al. entitled “White Light-Photolysis for the Removal of Polycyclic Aromatic Hydrocarbons from Proximity Firefighting Protective Clothing”.

In the manuscript Authors presented the investigations of the PAHs removal by white light photolysis (with or without the addition of hydrogen peroxide) from firefighting protective clothing, prepared in the laboratory.

The title reflects the content of the manuscript, the introduction part is informative.

Introduction:

Line 39: “PAHs consist of a class of organic materials with more than 100 compounds presenting in their structures two or more aromatic rings condensed in different ways.”

It would be better:

“PAHs  represent a class of complex organic chemicals with more than 100 compounds  containing two or more aromatic rings condensed in different ways, in their structures.”

Line 41: “PAHs can be formed by different pathways, as in an oil spill or oil seepage.”

Should be:

“PAHs can be formed by different pathways.” or “PAHs can be formed by different pathways, e.g. by pyrosynthesis.” (“oil spill or oil seepage” is  the way of PAHs dispersion – not formation).

In my opinion, the  introduction  should be rewritten in more organized form with use of the better English.

Others:

1.      Had the Authors check before starting the experiment, what specific PAHs and in what concentrations are usually adsorbed on firefighting protective clothing?

2.      Why have the Authors chosen 9-methylanthracene and pyrene for the tests and not any other PAHs? Why 0.005 M solutions has been used in the research? Do the concentrations of PAHs correspond to the actual  values?

3.      In the process presented by the Authors, oxygen derivatives of PAHs can be formed. However, they do not take up this issue in the discussion. It would be good to supplement the manuscript with the information if the  PAHs derivatives more harmful than the original PAHs haven’t been formed.

4.      Why the LOQ and LOD values for the PAHs examined in the work haven’t been calculated?

Minor flaws:

There is lack of references to: Figure S1, Figures S6-S30 and Tables 1-3, which are contained in the Supplementary material

Figure 3 and Figure S20 are the same, with slightly different captions. Is it a mistake or not?

All manuscript:  “decon” should be replaced with “decontamination

                  “PAH” should be “PAHs”

Lines 59 and 447: “physical decon techniques” should be “mechanical decontamination techniques”

Line 178: “All the data is available in the supplementary material” - What data? In what table or figure?

According to the above I recommend for this paper a major revision.

Author Response

We thank the referee’s suggestion and a point-by-point reponse is in the attachment.

Reviewer 2 Report

The manuscript entitled "White Light-Photolysis for the Removal of Polycyclic Aromatic Hydrocarbons from Proximity Firefighting Protective Clothing" from Arouca et al involved the evaluation of WLP for removing PAHs from proximity firefighting protective clothing. Moreover, the experiments were performed using H2O2 and time variation.

The introduction is according to the developed topic of the manuscript, and it has updated bibliographical references to support the research.

Also, the manuscript is interesting, clear, organize, and focused on the topic that is of interest due to its potential pharmaceutical uses and properties.

The specified methodologies are according to the aim of this research and the results are organized and widely discussed. 

Additionally, the information they described is supported with clear and logical images/figures/tables that summarize all the obtained information and data.

As a suggestion, it should be important to consider the standard deviation data in all the graphics and it is important to report the number of samples for each assay (n =……)

Considering the potential application of this system it would be interesting to add a small paragraph where the authors could explain the projection of this systems with an industrial porpoise (as an example, to apply these methodologies for the real amount of firefighting clothes in one day of work)

Finally, I would like to invite the authors to add the abbreviation list of words at the end of this manuscript.

I recommend the acceptance of this manuscript after the authors performed the suggested corrections/additions.

Author Response

We acknowledge all contributions and your suggestions were accepted and properly amended and highlighted in the text. Regarding the projection of our  system in an industrial scale, we will submit a new manuscript on this topic.

Reviewer 3 Report

Dear Authors, 

The manuscript presented for review concerns issues related to the problems with repeated use of firefighting clothing. It is an important topic because it concerns a significant improvement in the quality of work and life of people working every day in very hazardous conditions. The methodology for measuring the ageing of fabrics customarily used in firefighting clothing presented in the manuscript is overall correct. Tests for the degradation of PAHs with visible light are promising, although it seems that to be sure how it affects the structure of the fibres of the protective clothing, the measurement methodology should be extended. The article itself should be more of preliminary research. In addition, consideration should be given to possibly carrying out actual protective clothing ageing measurements. It doesn't just rely on other people's measurements. The research of which, differed in certain points from the research presented in this paper. A replacement method for the use of a relatively expensive photodegradation device is the microfedometry technique. It is a relatively young non-destructive technique. Perhaps it will also find an application here because it has recently been mainly adopted in the field of so-called heritage science. However, since the topic under study requires a multidisciplinary approach and analysis, I believe this type of research can be undertaken in the future. My comments concern the way of formulating conclusions. In my opinion, the number of tests performed and the duration of the test are not large enough to say with certainty that this method of cleaning firefighting clothes is superior to others. It is the only substantive objection concerning this work. The other side is the minor linguistic errors. Despite their presence, the manuscript reads well.

From the point of view of strengthening the presented conclusions, I think that the theoretical part should be expanded with additional information on what substances are present on firefighter suits and then what chemical compounds are usually obtained after their degradation and whether they are still treated as hazardous to health. Moreover, I think that some of the so-called supplementary materials should be included in the main part of the manuscript.

Yours faithfully, 

Reviewer

Author Response

We acknowledge all contributions and your suggestions were accepted and properly amended and highlighted in the text.

Reviewer 4 Report

Comments to the authors

Title: White Light-Photolysis for the Removal of Polycyclic Aromatic Hydrocarbons from Proximity Firefighting Protective Clothing

UseID: ijerph-1782937

The article is written well and is under the scope of the journal. However, the following mandatory revisions must be done before considering publication in International Journal of Environmental Research and Public Health

1.      While using hydrogen peroxide, how do you control reaction with water

2.      Show post removal of PAH

3.      Hydrogen peroxide has bleaching effect, I want to see the effect on the fibre

4.      Effect of reaction time must be indicated for PAH

5.      The introduction is a little bit must be compressed

Author Response

(The authors gave the same response as above.)

Round 2

Reviewer 4 Report

Not well anwered